# Implementation and User Satisfaction Analysis of an Electronic Medication Reconciliation Tool (ConciliaMed) in Patients Undergoing Elective Colorectal Surgery

**DOI:** 10.3390/healthcare13070778

**Published:** 2025-03-31

**Authors:** Pablo Ciudad-Gutiérrez, Paloma Suárez-Casillas, Ana Belén Guisado-Gil, Héctor Luis Acosta-García, Isabel Laura Campano-Pérez, Nieves Ramírez-Duque, Eva Rocío Alfaro-Lara

**Affiliations:** 1Department of Pharmacy, Virgen del Rocio University Hospital, 41013 Seville, Spain; pablocg95@gmail.com (P.C.-G.); palomasuacas@gmail.com (P.S.-C.); hectoracostagarcia.hag@gmail.com (H.L.A.-G.); laura.campano.sspa@juntadeandalucia.es (I.L.C.-P.); eralfarolara@gmail.com (E.R.A.-L.); 2Institute of Biomedicine of Seville, Virgen del Rocio University Hospital, CSIC, University of Seville, 41013 Seville, Spain; 3Department of Internal Medicine, Virgen del Rocio University Hospital, 41013 Seville, Spain; nievesramirezduque@yahoo.es

**Keywords:** digital health, medication reconciliation, perioperative care, general surgery, mobile applications

## Abstract

**Background/Objectives**: Medication reconciliation is an essential strategy to improve patient safety, especially in polymedicated and chronic patients undergoing surgery. This study describes the implementation of an electronic medication reconciliation tool, ConciliaMed, in a multidisciplinary medication reconciliation programme performed for patients undergoing elective surgery and assesses user satisfaction with the tool since its release. **Methods**: A prospective observational study was carried out on “high-risk” patients undergoing colorectal surgery. In the medication reconciliation programme, ConciliaMed was mainly used to obtain an optimised and reconciled patient medication list by using the “Perioperative medication reconciliation” and the “Therapeutics equivalents” modules included in the tool. Data were registered about the reconciled medications, medication discrepancies and interventions made to optimise the reconciled medication list. Concerning the users’ satisfaction analysis, data about users’ registration and feedback were collected. **Results**: Seventy-three patients were enrolled in this study who were mainly polymedicated. A 10.1% of medication discrepancies were identified from the total of 553 reconciled drugs. The pharmacotherapeutic groups most involved in medication discrepancies were psycholeptics or diuretics. Regarding the optimisation of the reconciled drugs, stopping medication before surgery was the most frequent recommendation provided by the tool. According to the results of the satisfaction surveys, high overall satisfaction with the tool (4.45 ± 0.80) was reported by users. **Conclusions**: Stopping medication before surgery was the most common preoperative medication management recommendation provided by the tool. ConciliaMed was evaluated by pharmacists, nurses and physicians who reported a high level of satisfaction with the tool. A more comprehensive evaluation of this tool in other types of scheduled surgical patients is expected.

## 1. Introduction

Today, the increasing number of elective surgeries worldwide is attributed to advances in surgical practices, perioperative monitoring and a greater life expectancy [1]. Approximately one-third of elective surgeries are performed on the elderly, who have a higher risk of mortality, morbidity, polypharmacy and postoperative complications [2]. This population is also considered vulnerable to medication safety errors, in particular, those related to medication discrepancies [3]. A prospective observational study showed that 13% of 455 patients undergoing orthopaedic elective surgery had unintentional medication discrepancies, which were mainly related to the omission or incorrect medication [4]. According to the Joint Commission [5] and the World Health Organization [6], medication reconciliation (MedRec) is the formal process of comparing the medications that the patient is actually taking with the new medications ordered for the patient and resolving any medication discrepancies detected. Therefore, it is an effective strategy to avoid medication errors and discrepancies and improve the quality of care [7]. Previous works have shown the positive impact of MedRec on health outcomes. A pre-post study showed that the implementation of a MedRec program at admission to a care centre reduced the 30-day readmission rate of patients by 29.7% [8]. Furthermore, a prospective study showed that the median length of stay of patients decreased from 9.6 to 8.9 days (*p* = 0.024), and the 30-day readmission rate was reduced from 7.8% to 4.8% (*p* = 0.046) after the implementation of a pharmacist-led MedRec program in a medical unit [9]. In the perioperative context, a pre-post quasi-experimental study showed that the implementation of a multidisciplinary MedRec program [10] in patients scheduled for colorectal surgery reported a positive impact on the length of stay of patients aged over 75 years old and those with cardiovascular disease.

To support the MedRec process, the use of digital health constitutes an innovative and potential tool to facilitate this process for healthcare professionals (HCPs), so the integration of electronic health tools into clinician workflow should be promoted in healthcare centres [11,12]. Digital health is recognised as an effective strategy to improve patient care and well-being through the appropriate use of technology [13]. In recent years, there has been an exponential growth of commercially available mobile health applications (m-health apps) in the markets [14]. In this scenario, an adequate assessment of these tools is essential to selecting a sophisticated app and achieving long-term user engagement [15]. Some authors [16] have recommended the use of validated scales such as the ’Health Information Technology Usability Evaluation Scale’ [17] and the ’Mobile App Rating Scale’ [18] to guide users in selecting the most appropriate app by assessing its content and effectiveness. Analysing users’ feedback on m-health apps could also be used to assess the quality of these tools, especially their design and usability [19]. However, a systematic review found that more efforts are needed to define quality assessment criteria for future stakeholders in order to facilitate the selection of high-quality m-health apps [20].

A low number of electronic MedRec tools [21,22] have been implemented in the perioperative environment to date. Regarding patient safety, a reduction in the percentage of medication discrepancies was detected with the use of the ‘Aplicon’ tool [21] in surgical patients compared with the non-use of the tool (6.6% vs. 10.6%) and as it was observed with the use of the ‘RightRx’ [22] tool (26.4% vs. 56%). Recently, an electronic MedRec tool aimed at all HCPs involved in the MedRec process, ConciliaMed [23], has been incorporated into a multidisciplinary MedRec program to reconcile the chronic medications of patients undergoing elective surgery. Interactive modules and evidence-based content, together with the generation of a reconciliation report, were some of the strategies included in the tool to save time and workload for HCPs and improve patient safety. Thus, the aim of this study was to evaluate the implementation of ConciliaMed as an m-health app in a MedRec program for patients undergoing elective colorectal surgery and assess user satisfaction with the tool since its release.

## 2. Materials and Methods

### 2.1. Implementation of ConciliaMed

#### 2.1.1. Setting and Study Design

This study was conducted at the Virgen del Rocio University Hospital in Seville (Spain). In 2017, a MedRec program was developed and implemented in the preoperative setting for patients undergoing elective colorectal surgery. The MedRec program involved the collaboration of pharmacists, surgeons and internists at this hospital. From 2021 to 2023, ConciliaMed went through the phases of preparation, design and development, testing and dissemination before its implementation in clinical practice. The website tool is freely available online (https://conciliamed.chronic-pharma.com, accessed on 1 May 2023) and the mobile app through the Google Store. This process has been described previously [23]. In May 2023, ConciliaMed was implemented as a MedRec tool in the multidisciplinary MedRec program. Therefore, we designed a prospective observational study from 1 May 2023 to 31 December 2024 to analyse the MedRec process with the use of ConciliaMed and evaluate the users’ satisfaction with the tool.

#### 2.1.2. Participants

Adult patients (≥18 years old) undergoing elective surgery for colorectal cancer and with at least one of the following ’high-risk’ criteria for postoperative complications were eligible for inclusion in the MedRec program:Polypharmacy, defined as the concomitant use of five or more chronic medications.Diabetes mellitus type 1 or 2.Advanced chronic conditions:c.1. Cardiovascular disease is defined as congestive heart failure with dyspnea on exertion or ischemic heart disease with acute coronary syndrome or undergoing catheterisation in the last 12 months.c.2. Chronic renal failure (filtration glomerular < 30 mL/min).c.3. Lung disease with dyspnea on exertion.c.4. Liver disease and cirrhosis (Child-Pugh B or C).c.5. Hypertension

ConciliaMed was used in all patients enrolled in the MedRec program during the period of the study.

#### 2.1.3. Intervention

A detailed description of the MedRec program has been published in a previous study [10]. This program started with the recruitment of ’high-risk’ patients scheduled for colorectal surgery one week prior to admission. Pharmacists then interviewed the patient/caregiver to find out the patient’s current medications and compared them with the medication list extracted from the electronic medical records (EMRs). Once the reconciled medication list was available, the pharmacists also optimised these medications to continue, discontinue or replace the patient’s chronic medications prior to surgery based on clinical recommendations obtained manually and the anaesthesia reports. A subsequent review of the pharmacists’ interventions was conducted by the surgeons and internists responsible for the patient’s follow-up until discharge.

A summary of the MedRec program after the implementation of ConciliaMed is shown in Figure 1. First, pharmacists entered the patient’s reconciled medication list into the tool, together with their allergies and intolerances. Then, a reconciliation report was generated in the “Perioperative medication reconciliation” module to provide recommendations to patients/caregivers about the preoperative management of the patient’s chronic medications in accordance with the ConciliaMed recommendations and the anaesthesia reports. In addition, the “Therapeutic Equivalents” module integrated into ConciliaMed was used to find a therapeutic equivalent for any of the patient chronic medication, including its dosage regimen and route of administration. Finally, the reconciliation report was saved in the “My reconciliation reports” module of ConciliaMed and a proposal of the reconciled and optimised medication list was made by the pharmacists for review by the internist and surgeon before the surgery.

#### 2.1.4. Data Collection and Study Measures

Data were collected from EMRs and from ConciliaMed.

We registered the sociodemographic variables related to the target intervention population: sex, age (≤75 or >75 years old), body mass index (BMI), colorectal tumour location (colon or rectum), American Society of Anesthesiologists (ASA) physical status classification, polypharmacy, diabetes, advanced chronic conditions and hypertension.

Regarding the MedRec process, we recorded the following variables:The total and the median number of reconciled medications per patient.The number and type of medication discrepancies between the medication list extracted from the EMRs and the reconciled medication list were recorded after the interview with patients or caregivers (discrepancies related to doses, frequency, route of administration, addition or discontinuation of medications). The percentage of patients with at least one discrepancy was recorded.Pharmacotherapeutic groups, according to the Anatomical Therapeutic Chemical Classification System (ATC) code, involved in the medication discrepancies and if there were any of these medications classified as “high-risk” according to the High-Alert Medications for patients with Chronic illnesses (HAMC) list [24].The number of interventions made optimising the reconciled medication list to the perioperative setting based on ConciliaMed and the anaesthesia reports related to stopping or replacing the patient’s chronic medication before surgery. The pharmacological groups commonly involved in these interventions were also evaluated.Conversed medications according to the equivalent doses of drugs included in the “Therapeutics equivalents” module of the tool.Number and type of changes related to dose, frequency, route of administration or discontinuation of medications made by physicians to the reconciled and optimised medication list proposed by pharmacists.

User satisfaction analysis

Data from users’ profiles were retrieved from ConciliaMed. We analysed the users’ feedback and their satisfaction with the tool through the results of the satisfaction survey in order to refine ConciliaMed in the next updated versions. The following outcomes related to the users and user satisfaction were recorded:The number of registered users in the app from the day the tool was released on Google Play Store (5 March 2024) to 31 December 2024. Sex, age and professional category of the users were also collected.The number of satisfaction surveys completed by users.The means score of a 7-question survey [Figure 2]. A Likert-type scale was used to rate each question from 1 to 5.Users’ comments to suggest changes to improve the app’s usability.

#### 2.1.5. Statistical Analysis

Statistical Package for the Social Sciences (SPSS) version 23^®^ was used for analyses. Descriptive measures of the variables were calculated: categorical variables were presented as frequency distribution and percentages, and continuous variables were presented as means ± standard deviations (SDs) or medians and interquartile ranges (IQRs).

#### 2.1.6. Funding and Ethics Approval

This work was supported by the Instituto de Salud Carlos III, the Spanish Ministry of Science and Innovation (DTS20/00052), and was partially funded by the European Development Regional Fund “A Way to Achieve Europe”. This study was conducted according to the guidelines of the Declaration of Helsinki and Good Clinical Practice guidelines and approved by the Institutional Review Board (or Ethics Committee) from the Virgen del Rocio University Hospital and Virgen Macarena University Hospital No. 1089-N-21 (approved on 1 November 2021).

## 3. Results

### 3.1. Implementation of ConciliaMed

Table 1 shows the characteristics of the 73 patients recruited in the study. Most patients enrolled were men >75 years old and diagnosed with colon cancer. Concerning the ASA physical status, they were commonly classified as ASA III. We found that the main “high-risk” criteria for inclusion were polypharmacy followed by diabetes. Cardiovascular and renal disease were the most prevalent advanced chronic conditions. Also, the majority of patients presented with hypertension.

The total number of reconciled drugs after the interview is 553, and the median (IQR) of reconciled drugs per patient is 7 (4–20). Fifty-six (10.1%) medication discrepancies were detected from the total number of reconciled drugs: they were related to medications that were discontinued (n = 49), changed in doses (n = 4) or frequency (n = 3). There were no discrepancies regarding the route of administration and addition of new treatments. A total of 31 (42.4%) patients presented at least one discrepancy between home medication and EMRs: 21 (67.7%) patients presented one discrepancy, 6 (19.4%) patients presented two discrepancies and 4 (12.9%) patients presented three or more discrepancies.

Psycholeptics (N05), diuretics (C03) and blood substitutes and perfusion solutions (B05) were the pharmacotherapeutic groups most involved in the medication discrepancies, with seven, five and five discrepancies, respectively. A total of 19 (33.9%) “high-risk” medications, according to the HAMC list, were identified in the 56 medication discrepancies detected. Specifically, furosemide and bisoprolol were the most common medications associated with these discrepancies, followed by acetylsalicylic acid and ibuprofen. Detailed data are included in Table 2.

A total of 130 (23.5%) medications were optimised from the total number of reconciled drugs. Ninety-three (71.5%) medications were stopped before surgery. This involved the suspension of oral antidiabetics (OADs) (41.9%), antiplatelet agents (18.3%), oral anticoagulants (14%), antigout agents (11.8%) and the remaining percentage was represented by other drugs, such as pentoxifylline or naproxen (14%). Subcutaneous insulin was introduced in 46.2% of patients who suspended the treatment with OADs. Thirty-seven (28.5%) of the optimised medications were changed before the surgery. This included the replacement of clopidogrel with acetylsalicylic acid (AAS) in 2 (5.4%) patients, and in 1 (2.7%) patient, the dose of AAS was reduced. It also included the omission of diuretics in the morning of the day of surgery in 28 (75.7%) patients and the switching of proton bomb inhibitors in 6 (16.2%) patients according to the information contained in the “Therapeutics equivalents” module.

No changes were made to the reconciled and optimised medication list by internists or surgeons in 58 (83.6%) patients. Twelve (16.4%) patients presented at least 1 change: six (8.2%) patients presented 1 change, three (4.1%) patients presented 2 changes, and three (4.1%) patients presented 3 or more changes. They made a total number of 21 changes in the medication list proposed by the pharmacist. These changes were related to different doses (n = 6), frequency (n = 1) or suspended medications (n = 14).

### 3.2. User Satisfaction Analysis

There were 352 users registered in the app during the study period, 242 (68.8%) of whom were women. The median (IQR) age of the users was 39 years. Regarding profession, 293 (83.3%) were pharmacists, including hospital pharmacists (n = 276) and primary care pharmacists (n = 15). The remaining users were 10 (2.8%) nurses and 49 (13.9%) physicians, including anaesthetists (n = 18), general practitioners (n = 11), surgeons (n = 9), internists (n = 8), geriatricians (n = 2) and one medical student (n = 1). Table 3 shows the results of the 42 satisfaction surveys completed by the users. Overall, users reported a high level of satisfaction with the tool, with a mean score of (4.45 ± 0.80) in this item. The organisation and content of the tool were one of the items with a higher mean score (4.45 ± 0.89), while the design and visualisation of the app had the lowest score (4.28 ± 1.04). The inclusion of information about certain drugs in which perioperative management recommendations were lacking, and the integration of new functionalities into the web platform were some of the suggestions made by the users. Consequently, the researchers added more content to the tool by exploring new perioperative medication management guidelines [25] and agreed that the information from the “Perioperative medication reconciliation” module would be incorporated into the web platform in the near future.

## 4. Discussion

We describe the implementation of ConciliaMed in a multidisciplinary MedRec program for patients undergoing elective colorectal surgery in a tertiary care hospital. Seventy-three patients were included in the study, and a total of 553 drugs were reconciled by the pharmacists. The number of optimised medications was 130, which were mainly suspended prior to surgery according to the recommendations provided by the app and the anaesthesia reports. No changes were made by physicians in most of the reconciled and optimised medications. Regarding the user satisfaction analysis, 352 healthcare professionals were registered in the app and 42 of them completed the satisfaction survey. All the items were rated positively by the users. The feedback from the users allows the researchers to consider the incorporation of new functionalities in the next updates of the tool to meet some of the users’ needs.

Some authors have suggested that polypharmacy, along with both older age and the presence of chronic conditions, are some of the patient risk factors associated with a higher percentage of medication discrepancies [26]. This could explain the higher percentage of patients with at least one medication discrepancy found in this study (42.4%) compared with the 38.1% found in the implementation of another electronic MedRec tool, Aplicon [21], in a general surgery department. Regarding the categorisation of these discrepancies, a previous study [27] conducted in an internal medicine ward showed that omission of medication was the most common discrepancy detected, followed by incorrect dose, as observed in our results. Additionally, the pharmacotherapeutic groups most associated with medication discrepancies were cardiovascular agents, which is consistent with the high proportion of diuretics and beta blockers involved in the medication discrepancies observed in our study.

Han et al. [28] showed the most common pharmacist interventions related to medication management in the perioperative setting. A significant number of dose changes were recommended by the pharmacists, particularly for tapering of benzodiazepines and estrogen-containing therapies prior to surgery. This contrasts with the results of our work, which showed that only one dose change was recommended in the preoperative setting. These authors also mentioned that appropriate management of oral anticoagulants is essential in surgical patients, which is consistent with the high percentage of preoperative recommendations related to oral anticoagulants performed in our study. Furthermore, a systematic review [29] reported that pharmacists’ recommendations, including the optimisation of reconciled medications, were widely accepted by physicians (77% to 97.1%) in the perioperative setting. A low number of changes in the proposed list by the pharmacists was achieved with the use of ConciliaMed, which may be influenced by the generation of a reconciliation report with updated and reliable perioperative medication management recommendations.

A previous systematic review analysed users’ opinions of available electronic MedRec tools [30]. Users of the ’Twinlist’ tool [31] indicated that feedback from users was needed to guide the implementation of new features in the next updates of the tool. Following this recommendation, registration with ConciliaMed via the website is now available to all users, providing an alternative option for users with iOS systems-included phones. Improving the workflow integration of some electronic MedRec tools [32,33] was mentioned by some HCPs who reported a lack of usability of these tools in the MedRec process. In this sense, intuitive and easy-to-use modules were included in ConciliaMed to speed up the MedRec process and minimise the HCPs’ workload. For example, the “Frequently Asked Questions” module was reported by some HCPs to be particularly useful in resolving their queries about the navigation or content included in the tool. These functionalities may have contributed to the widespread use of ConciliaMed across different professional categories, such as pharmacists, physicians and nurses, who reported high overall satisfaction with the tool.

### Limitations of the Study

This study reports the results of the use of ConciliaMed in patients undergoing elective colorectal surgery in a tertiary care hospital. It describes the implementation of ConciliaMed and users’ satisfaction analysis. However, clinical and economic parameters could be evaluated in future work to assess the impact of the tool on health outcomes. Although the use of ConciliaMed could be extended to other elective surgical procedures and/or other healthcare centres, the information contained in this study should be extrapolated with caution because the implementation of ConciliaMed was made within a specific multidisciplinary MedRec program. In addition, the implementation of ConciliaMed was carried out exclusively in the preoperative context, but this tool could support HCPs throughout the perioperative period. In addition, only 42 of the 352 registered users completed the satisfaction survey. A high average satisfaction rate was obtained from the respondents to the survey, but a higher response rate is needed to reflect the opinion of all users registered in the tool.

## 5. Conclusions

This study presents the implementation of an electronic MedRec tool, ConciliaMed, in a MedRec program in order to optimise the patient’s chronic medication list prior to elective colorectal surgery. Stopping medication before surgery was the most common preoperative medication management recommendation provided by the app, especially for OADs and antiplatelet agents. A low percentage of changes was made by physicians to the reconciled and optimised medication list proposed by the pharmacist. ConciliaMed was disseminated among pharmacists, nurses and physicians who reported a high level of satisfaction with the tool.

## Figures and Tables

**Figure 1 healthcare-13-00778-f001:**
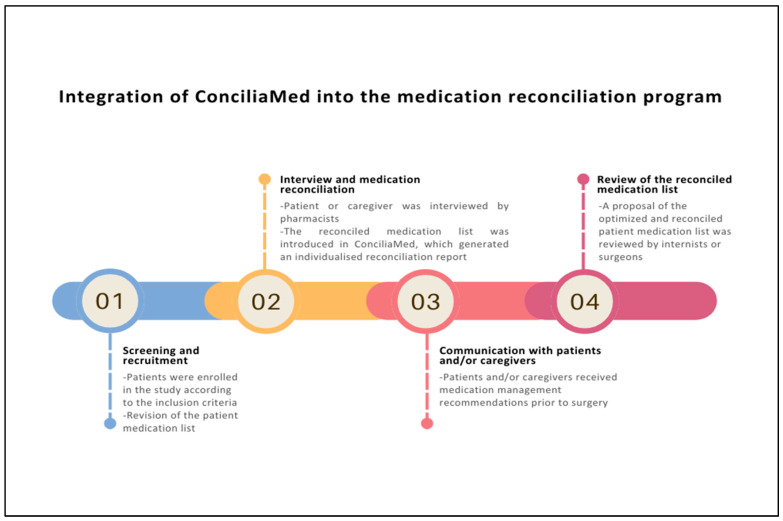
Integration of ConciliaMed into the medication reconciliation program.

**Figure 2 healthcare-13-00778-f002:**
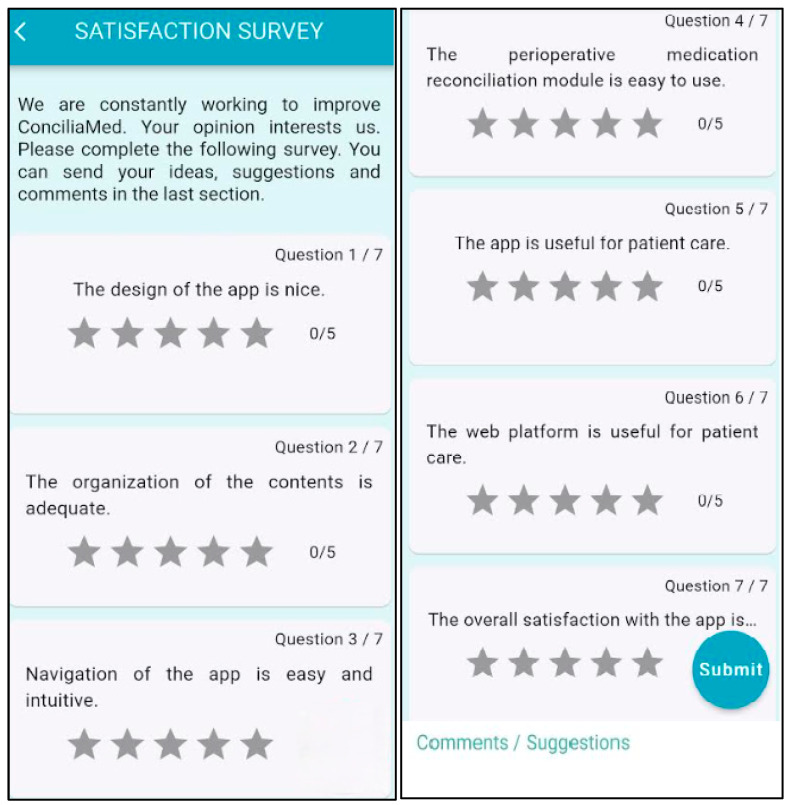
Satisfaction survey.

**Table 1 healthcare-13-00778-t001:** Baseline characteristics of patients.

Characteristics	Intervention Group n = 73
Sex, n (%)	
Male	56 (76.7)
Female	17 (23.3)
Age	
≤75 years, n (%)	36 (49.3)
>75 years, n (%)	37 (50.7)
Body mass index, mean ± SD ^a^	29.3 ± 5.1
Primary tumor location, n (%)	
Colon	61 (83.6)
Rectum	12 (16.4)
ASA ^b^ physical status classification, n (%)	
ASA I	2 (2.7)
ASA II	22 (30.1)
ASA III	49 (67.1)
Polypharmacy, n (%)	57 (78.1)
Diabetes, n (%)	35 (47.9)
Advanced chronic conditions, n (%)	14 (19.7)
Cardiovascular disease	8 (8.7)
Lung disease	0 (0)
Liver disease	2 (2.7)
Kidney disease	4 (5.5)
Hypertension, n (%)	55 (76.0)

^a^ SD, standard deviation; ^b^ ASA, American Society of Anesthesiologists.

**Table 2 healthcare-13-00778-t002:** Pharmacotherapeutic groups by ATC ^a^ code and “high-risk” ^b^ medications involved in the medication discrepancies.

**Medication discrepancies (n = 56)**	**Pharmacotherapeutic Groups (n)**	**“High-Risk” Medications (n)**
	A02. Drugs for acid-related disorders (2)	
	A06. Drugs for constipation (4)	
	A12. Mineral supplements (1)	
	B01. Antithrombotic agents (3)	Acetylsalicylic acid (2) Enoxaparin (1)
	B05. Blood substitutes and perfusion solutions (5)	
	C03. Diuretics (5)	Furosemide (3) Spironolactone (1)
	C05. Vasoprotectives (1)	
	C07. Beta blocking agents (4)	Bisoprolol (3) Atenolol (1)
	C08. Calcium channel blockers (1)	
	C09. Agents acting on the renin-angiotensin system (3)	
	C10. Lipid modifying agents (4)	
	J01. Antibacterials for systemic use (2)	
	M01. Anti-inflamatory and antirheumatic products (2)	Ibuprofen (2)
	N02. Analgesics (3)	Tramadol (1)
	N05. Psycholeptics (7)	Lorazepam (2) Diazepam (1) Alprazolam (1) Risperidone (1)
	N06. Psychoanaleptics (3)	
	R03. Drugs for obstructive airway diseases (3)	
	R05. Cough and cold preparations (1)	
	R06. Antihistamines for systemic use (2)	

^a^ ATC, Anatomical Therapeutic Chemical Classification. ^b^ “High-risk” medications according to the High-Alert Medications for Patients with Chronic Illnesses (HAMC) list [20].

**Table 3 healthcare-13-00778-t003:** User’s satisfaction analysis and next updates of ConciliaMed.

Items Evaluated	Mean Score ± SD ^1^ [Score: 1–5]	User’s Comments	Next Updates of ConciliaMed
Design and visualisation	4.28 ± 1.04	“New professional categories should be included in the tool to be selected during the registration process, for example, community pharmacists” “The FAQs module was useful in enabling direct communication with researchers in order to quickly resolve questions about the tool’s navigation or content”	More professional categories will be added in ConciliaMed according to the demand from healthcare professionals.
Organisation and content	4.45 ± 0.89	“It is desirable to expand the information about certain drugs which perioperative management recommendations are missing” “The guidelines about perioperative medication management included in the “Documents of interest” module was useful to consult some queries about certain information of medication management”	New sources of information are explored periodically by researchers to provide more information about the perioperative management of medications with limited data, such as monoclonal antibodies or immunosuppressants.
Navigation	4.40 ± 0.91	“It would be interesting to integrate a scanner into the app to automatically extract the information about the patient medication list from electronic health records”	Automatic data entry into the tool will be evaluated by researchers in the near future.
Easy-to-use of the perioperative medication management module	4.38 ± 0.91	“It would be necessary to select automatically an individualized perioperative management recommendation in certain medications, for example, DOACs, with a different perioperative management according to the type of surgery or patient risk factors”	A free text option is provided to be completed by healthcare professionals, especially for those medications with high variability in the perioperative management.
App usability	4.31 ± 0.87	“The development of an iOS version would enhance the rapid spread of the app”	ConciliaMed is developed for Android systems, but all users can also be registered via the web platform. The development of an iOS version will be taken into account by researchers in the next year.
Web platform usability	4.35 ± 0.93	“It would be helpful for users that the generation of reconciliation reports could be also made through the web platform”	All users will have access to the information included in the “Perioperative medication management” module in the web platform, which was not implemented up to date. The generation of reconciliation reports in the web platform will be explored by researchers in the future according to healthcare professionals’ needs.
Global satisfaction	4.45 ± 0.80	“It is an easy-to-use and intuitive tool to carry out the MedRec process, especially for those healthcare professionals who are working in the perioperative setting”	

^1^ SD: standard deviation; App: application; FAQs: frequently asked questions; DOACs: direct oral anticoagulants; MedRec: medication reconciliation.

## Data Availability

The data shown in this study can be requested from the corresponding author upon reasonable request.

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
