# Peer review of "Implementation and User Satisfaction Analysis of an Electronic Medication Reconciliation Tool (ConciliaMed) in Patients Undergoing Elective Colorectal Surgery"

_healthcare, 2025, doi:10.3390/healthcare13070778_

Round 1
Reviewer 1 Report
Comments and Suggestions for Authors
I read with interest the paper titled "Implementation and user satisfaction analysis of an electronic medication reconciliation tool (ConciliaMed) in patients undergoing elective colorectal surgery"
I have some comments that could be used to improve the manuscript.
1. An Up to date definition of Medication Reconciliation should be added in the background.
2. Regarding the presentation of information, sometimes, it's not quite clear to me what is a new paragraph or what is a bullet point arising from previous indication. As an example, information between line 143 to 162, could be presented as (1) bullet points; or (2) a table. The same happens between 169 and 175. The use of advance spaces make the paper not very clear in that sections.
3. How was the sample os 73 calculated? Did the authors enrolled and analysed all patients that entered in Med Rec process during the period of the study, or a sampling technique was applied?
4. Somewhere in the manuscript, the users of the ConciliaMed should be clearly defined. Who are they? Pharmacists, nurses, practitioners? In line 89-90 you define who is doing MedRec in the hospital. Are them the users?
Reviewer 2 Report
Comments and Suggestions for Authors
The paper describes a study evaluating the implementation of an electronic medication reconciliation programme and satisfaction among patients for that. The authors have assessed, as outcomes, the therapy related recommendations by the programme such as stopping medication before surgery. Various drug related factors have been analysed like medications reconciled, discrepancies etc. Satisfaction analysis among users was also done.
What was the impact of the intervention, however, on clinical parameters? The authors could have incorporated the clinical outcomes also among various others such as duration of hospitalisation, drug related adverse events, readmission after initial hospital discharge etc. Also, the economic impact could also have been assessed in terms of direct and indirect costs, additional costs saved while using the software etc.
Introduction: The authors mention “Previous works have shown the positive impact of MedRec on health outcomes”. Here, a brief account of the existing literature on this regarding the conditions where it has been tested and assessed health outcomes may be given.
A brief description of the similar existing tools and their applications may be added in the introduction.
For user satisfaction, the authors have used a 7-question survey; a brief description of the questionnaire including the type and content of questions may be added for a better understanding of the readers. Also, how was the aggregate score related to satisfaction.
Line no. 327 states “the high mean scores obtained in all the items may reflect the opinion of all users registered in the tool”; this may not be true as the number of responders was much less than the registered users. In-fact, such low rate may also be due to less satisfaction among users.
Reviewer 3 Report
Comments and Suggestions for Authors
The dates within the references are missing dates in couple of place(eg: Reference #2,8,9 etc. Please ensure the format is consistent. Either you include day, month and year for all or only year and month.
Line 52: Could you please name the organizations? Because I checked one of them is reference of WHO. It would be more effective if you name them here.
Line 61: Word seems to be missing after easily, perhaps "accessible".
Introduction section, paragraph 1 (Line 53): In this section could you explain in a bit detail what is Med Rec and what it entails?
Introduction section , paragraph 2(Line 60-75): While authors have explained importance of digital health in great detail, it would be helpful if its usage with MedRec is explained more here.
Line 76, explain Concilia Med more here.
Section 2.2, Line 98: Define adults age criteria for inclusion,
Line 194: explain acronym ASA
Comments on the Quality of English Language
Proof reading is needed on the manuscript as at some places words seems to be missing.
Round 2
Reviewer 2 Report
Comments and Suggestions for Authors
No further queries from my side.